# Customers as Co-Creators: Antecedents of Customer Participation in Online Virtual Communities

**DOI:** 10.3390/ijerph16244998

**Published:** 2019-12-09

**Authors:** Yang Yang, Zhongqiu Li, Yingying Su, Shanshan Wu, Boyou Li

**Affiliations:** School of Management, Harbin Institute of Technology, Harbin 150001, China; yfield@hit.edu.cn (Y.Y.); 18b910059@stu.hit.edu.cn (Y.S.); lifeng19640605@126.com (S.W.); wangsuli19620224@163.com (B.L.)

**Keywords:** customer participation, perceived ease of use, perceived control, online virtual community

## Abstract

The development of internet technology and the popularity of smartphones has been gradually affecting people’s daily lives, thus causing subtle changes to their health. Manufacturing companies are increasingly establishing virtual communities to motivate customers to participate in new product development. However, the reasons that customers participate in the innovation process and the timing of participation remain under-researched. Hence, using data on 517 customers of 14 manufacturing enterprises, we investigate the reasons behind such participation and the moderating role of perceived ease of use and perceived control based on the social exchange theory. Results show that learning benefits, integrative benefits, and hedonic benefits have positive effects on participation. Perceived ease of use strengthens the positive relationship between integrative benefits and customer participation. Perceived control strengthens the positive relationship between hedonic benefits and customer participation. Theoretical implications and managerial practices are also discussed.

## 1. Introduction

The advancement of internet and smartphone technologies has led to people using these more frequently; hence, their daily life and mental health are gradually affected [1]. The same is true as regards consumers’ physical health. Considering the importance of mental health [2], it is necessary to have in-depth understanding of the factors motivating people to participate in virtual communities. Customer participation is important for enterprises and customers. For enterprises, customer participation can help shorten development time, control costs, and reduce uncertainty in the innovation process [3,4]. For customers, participation can improve their psychological interests [5]. In the process of interaction, the participating customers adhere to the principles of mutual dependence and mutual benefit and actively exchange information, knowledge, emotions and other resources [6]. Participating in innovation can help meet customer expectations and preferences, and it can also have a profound impact on their psychological and behavioral outcomes [5]. Therefore, further understanding of customer participation is crucial to enterprises and to the mental health of customers.

Customer participation in the innovation of manufacturing enterprises in virtual communities differs from traditional methods, such as offline experience shopping, face-to-face exchanges, user visits, and communication through emails and fax messages. Online communities provide online forums and virtual design services that enable customers to easily participate in design, testing, and support activities related to products [7]. Further, virtual communities use advanced technology, which ensures more real-time, personalized, and diverse business and customer exchanges, all of which can stimulate customers to participate in a company’s product planning, development, and marketing activities [8]. The increase in interaction between customers and enterprises gives customers a “sense of empowerment”, which further enhances their willingness to participate in creative activities [9]. Online communities create a free exchange platform for companies and customers to share information and help customers to participate in value co-creation [10]. More than 1.5 million enterprises have launched virtual communities for their brands, taking full advantage of the potential of social sharing to jointly create market value [11]. Therefore, online virtual communities have become an engine for customers and enterprises to co-create value [3].

Considering the complexity of customer participation in new product development, the literature ignores the important influence of multiple factors of customer-perceived benefits on such participation. Some scholars seek to analyze the antecedents of customer participation from the perspective of expected benefits of innovation [7]. However, others seek to do so from the perspective of self-confidence [12], locus of control [13], overall level of happiness [12], and the ability to contribute perspective, such as through their knowledge, skills, resource competence, and experience, which are considered antecedents of customer participation [14]. Although motivation triggers customer participation in value co-creation, empirical studies on motivation factors for customer participation in co-creation are limited [15].

Interestingly, the lack of research from a marketing as well as an empirical perspective has led to lack of focus on the consumer psychology of online community participation as well as active contribution. Thus, the objective of this study is to propose and evaluate an integrated model of online community participation to examine the dynamics of the relationships as well as the relationship between participation antecedents and customer participation. This study uses a framework of online community member benefits of the “uses and gratifications” approach [16]. In this framework, the customer gains from participation in virtual customer environments owing to cognitive, social integrative, personal integrative, and hedonic benefits. In this study, we focus on the antecedents of customer participation emerging from cognitive, integrative, and hedonic benefits. We consider two components of integrative benefits: personal and social integrative benefits. Previous studies have not systematically analyzed the ways to improve customer participation in online virtual communities, which has become a fundamental issue in operating the current communities. This study fills this gap by introducing customer-perceived benefits into the uses and gratifications approach [16]. From the perspective of these benefits, it explores factors that influence the emergence of customer participation in virtual communities, which support organizations in developing more advanced positioning tactics.

We discuss drivers of customer participation in new product development based on the social exchange theory, which is an important theory for analyzing interpersonal psychology and behavior [17]. Customer participation in innovation meets individual needs, both material and emotional. Individual needs are the premise on which exchange behavior is based [18]. Customers who have high levels of cognitive or learning benefits would need a solution to problems related to products and underlying technologies and their usage [7]. An online virtual community holds valuable collective knowledge regarding products and their use through continuous customer interaction [19]. According to the social exchange theory, a customer who expects that participation will provide certain specific benefits and will be motivated to participate in innovation. Further, we attempt to explain the relationships between other motivations and customer participation by applying this theory.

The benefits of customer participation in innovation to manufacturing enterprises are obvious, and the next questions regarding this are as follows: Why do customers participate in innovation? How can customer participation be increased? Does perceived ease of use and perceived control influence customer participation? It is necessary to identify the antecedents of customer participation in innovation within the context of virtual communities, that is, the expected or realistic benefits of customer participation, as well as the internal relationship between customer participation antecedents and participation? In view of these questions, this study investigates the relationships between customer participation antecedents and customer participation, as well as the moderating role of perceived ease of use and perceived control on such participation.

In the following sections, first, we develop the relationship between the antecedents of customer participation and customer participation by drawing on the social exchange theory. Second, we argue that this relationship is stronger in environments that have higher levels of perceived ease of use. Third, we investigate whether the relationship between antecedents of customer participation and customer participation is moderated through perceived control. The research model is depicted in Figure 1. Next, we test the research model by using survey data from 517 customers of 14 manufacturing enterprises in China. We conclude by describing the theoretical and practical contribution of this study to customer participation and social exchange theory-related literature.

## 2. Theory and Hypotheses

There are three perspectives on customer participation motivation: the individual, group, and social relationship theory; self-construction theory; and utility and satisfaction theory. Regarding the utility and satisfaction theory, Katz et al. was the first to propose four customer participation motivations: cognitive or learning benefits, social integrative benefits, personal integrative benefits, and hedonic benefits [16]. This framework has been widely used [20]. Wang and Fesenmaier study such motivation of the online travel community and suggest that it includes functional benefits, social interests, psychological benefits, and hedonic benefits [21]. Nambisan and Baron posit that the antecedents of customer participation are learning benefits, social integrative benefits, personal integrative benefits, and hedonic benefits [7]. Overall, the most widely cited customer participation antecedents are learning benefits, integrative benefits, and hedonic benefits. On studying customer participation in new product development, Kaulio suggests that such participation consists of vertical and horizontal dimensions [22]. On this basis, Fang explicitly states that customer participation refers to the breadth and depth of customer participation in new product development [23].

### 2.1. Participation Motivation and Customer Participation

Cognitive or learning benefits refer to the acquisition and enhancement of information and the understanding of an environment [16], reflecting customers’ understanding and learning about products, their underlying technologies, and their usage [7].

We believe learning benefits influence customer participation. First, the individual’s cognitive factor has a significant impact on individual behavior [24]. The higher the customer’s learning benefits, the more product information the customer will require. A virtual community provides customers a platform for sharing information and resources. Customer participation in innovation provides customers an opportunity of increased understanding of product knowledge, the latest development of enterprises and their products and technologies [7,25], discount information, and answers to questions related to products or an opportunity to find some information [21,26,27]. According to the social exchange theory, social interaction is essentially a two-way process of reciprocity [17]. Some customers participate to gain preferential information, learn, and accelerate decision-making [21]. Thus, we propose the following hypothesis:

**Hypothesis** **1.***Learning benefits are positively related to customer participation*.

In this study, integrative benefits consist of personal and social integrative benefits. Personal integrative benefits refer to the authority, reputation, position, and sense of achievement and self-confidence that customers gain through their participation [7,16]. Virtual communities serve as a venue for individual customers to exhibit their product-related knowledge and problem-solving skills. Social integrative benefits refer to the benefits that customers gain from social and interpersonal relationships, such as a sense of belonging, social identity [20,28], friendship, and social help [26], which they attain by building relationships with other customers and businesses as participation time grows [16].

First, in the context of a virtual community, rapid and frequent interactions and feedback create an environment of active communication among members and enterprises [25]. According to the social exchange theory, positive and beneficial behaviors from organizations and others contribute to establish high-quality exchange relationships and encourage individuals to engage in reciprocal acts [29]. In a positive communication environment, customers perceive that other customers or enterprises care for them and really understand their demands, thus promoting mutual understanding and social identity [7]. In mutual understanding and identification, community members gradually build up close interpersonal relationships, shared values, and a sense of belonging. As a result, customer participation will expand.

Second, customers can improve their community status and reputation through participation in innovation and gain abilities related to solving problems, providing information, and demonstrating product-related knowledge in the virtual community [19,21,26]. Virtual communities provide customers a platform to demonstrate product knowledge and skills. The social exchange theory holds that social exchange relationships are built over a long term and involve fewer tangible resources but more symbolic or social emotional resources, such as the exchange of recognition or self-esteem [17]. Helping others solve problems can allow customers to obtain corresponding personal integrative benefits in the group [30], making others think that they are talented [31,32].

Moreover, to maintain their reputation, customers will continue to participate and interact. They can achieve self-efficacy and satisfaction, owing to their participation in innovation, by influencing other people’s use of products and improving enterprise product plans [28], making customers believe that they are valuable and competent [33]. The enhancement of self-efficacy makes them willing to do more. Third, the acquisition of reputation and self-efficacy takes time to accumulate. Customers need to continue showing their professional knowledge and to constantly receive feedback. Feedback is an ongoing process. After a customer’s solution is applied to the product environment, it is possible to identify the solution’s effect [7]. In this process, to ensure their solution is effective, customers will continuously participate in new product development. Therefore, this study proposes the following hypothesis:

**Hypothesis** **2.***Integrative benefits are positively related to customer participation*.

Hedonic benefits refer to the benefits that a customer gains through participation, such as aesthetic and spiritual pleasure, fulfilment of interests, pleasant experiences. and entertainment [7,16].

Some customers might find it fun to communicate with others about product-related information [34]. A virtual community’s relaxing network environment presents customers an opportunity to explore new worlds full of fantasy and entertainment [21]. Simultaneously, rapid, frequent interaction on the internet also creates a more aware, attractive online environment for customers [35]. The social exchange theory posits that people exchange six types of resources: emotion, status, information, money, goods, and services [36]. They can participate in innovation because of their emotional needs, which involves the exchange of emotion and information. The direct perception of pleasure or the pleasure gained from one act leads to an expansion of participation [37]. In the process of solving problems and generating ideas, they can activate their thinking, obtain spiritual stimulation and intellectual stimulation, and satisfy hedonic demands [21], which will encourage them to participate in more stages to obtain fun and pleasure. Therefore, this study proposes the following hypothesis:

**Hypothesis** **3.***Hedonic benefits are positively related to customer participation*.

### 2.2. Perceived Ease of Use and Customer Participation

Davis believes that perceived ease of use, based on the perceived ease of engagement, can affect customers’ attitude towards technology innovation. Perceived ease of use refers to the degree to which users perceive the use of a particular system as being effortless. In other words, the system is not difficult to use, or it does not require great effort [38]. This theory has been widely applied to study the relationship between computer information technology and customer adoption behavior. Perceived ease of use will affect customer attitudes towards new technology, thereby further affecting their behavioral intentions [39]. Behavioral intention directly affects the innovation behavior of customers [40]. This study considers that customers have many avenues to participate, such as face-to-face visits, customer meetings, customer observation and feedback, exhibition, experience centers, and online virtual communities [41]. Different customer participation patterns yield different levels of perceived ease of use, thus affecting the behavior of customer participation.

First, a virtual community provides great convenience for customer participation. It provides a platform for customers to obtain information, stay connected with others, deepen relationships, and befriend like-minded people [21]. From the perspective of social exchange, they will rationally calculate the costs of participation in innovation and comply with the norms of fair exchange [42]. The higher the perceived ease of use, the more likely customers are to consider the exchange worthwhile, and the more likely they are to participate rationally in other stages of innovation. Second, a customer’s interface style and interaction will affect their sense of computer self-efficacy and, thus, affect their perceived ease of use. Fun interface styles can make customer experiences more enjoyable and can serve as an intrinsic motivation for stimulating behavior. Interactivity can deepen interpersonal relationships among members, and convenient access can encourage participants to maintain closer relationships with others in a wide scope of activities [43]. Therefore, this study proposes the following hypothesis:

**Hypothesis** **4.***Perceived ease of use is positively related to customer participation*.

### 2.3. The Moderating Effect of Perceived Ease of Use

We argue that perceived ease of use can serve as a moderator in the relationship between antecedents of customer participation and customer participation.

First, when perceived ease of use is high, it is easier for customers to communicate [43], which effectively facilitates the exchange of knowledge within the virtual community. According to the principle of reciprocity in the social exchange theory, balanced reciprocity entails timely, equal returns [44]. When the customer knowledge exchange process is timely and effective, thus meeting the principle of balanced reciprocity, customer participation based on the three types of motivations is more likely occur [44]. Second, a high level of perceived ease of use can yield customers the expected results, which will promote future customer innovation [37]. According to the social exchange theory, individuals are more likely to engage in social exchange in a conducive and effective exchange environment [17]. When the perceived ease of use level is low, customers perceive that participation will require greater efforts; hence, exchange is more difficult to produce [17], and the relationship between customer participation and customer motivation is hindered. Therefore, this study proposes the following hypotheses:

**Hypothesis** **5.***Perceived ease of use moderates the relationship between antecedent of customer participation and customer participation*.

**Hypothesis** **5a.***Perceived ease of use moderates the relationship between learning benefits and customer participation. Specifically, the effect of learning benefits on customer participation is stronger when there are high levels of perceived ease of use and weaker otherwise*.

**Hypothesis** **5b.***Perceived ease of use will moderate the relation between integrative benefits and customer participation. Specifically, high perceived ease of use will enhance the effect of integrative benefits on customer participation, and low perceived ease of use will diminish the effect*.

**Hypothesis** **5c.***Perceived ease of use will moderate the relation between hedonic benefits and customer participation. Specifically, high perceived ease of use will enhance the effect of hedonic benefits on customer participation, and low perceived ease of use will diminish the effect*.

### 2.4. The Moderating Effect of Perceived Control

We propose that perceived control can serve as a moderator in the relationship between antecedents of customer participation and customer participation. Perceived control focuses on the individual’s gains from controlling the surrounding environment [45]. People’s perception of control can affect their attitudes and behavioral intentions [46,47]. Individual behavioral intentions are enhanced by an increased level of control over their surroundings. Perceived control allows customers to define a series of steps that the new product development team follows, to monitor the extent to which the team actually follows established procedures, and, lastly, to evaluate the process used to accomplish a given task. Appropriate controls can help customers and manufacturers achieve mutual understanding of customer needs and expectations, and establishing a close relationship between customers and manufacturers is the key to the success of new products. Employees with high levels of perceived control may consider that their participation will be more meaningful than those with low levels of perceived control [13]. Therefore, this study proposes the following hypotheses:

**Hypothesis** **6.***Perceived control moderates the relationship between antecedent of customer participation and customer participation*.

**Hypothesis** **6a.***Perceived control moderates the relationship between learning benefits and customer participation. Specifically, the effect of learning benefits on customer participation is stronger when there are high levels of perceived control and weaker otherwise*.

**Hypothesis** **6b.***Perceived control will moderate the relation between integrative benefits and customer participation. Specifically, high perceived control will enhance the effect of integrative benefits on customer participation, and low perceived control will diminish the effect*.

**Hypothesis** **6c.***Perceived control will moderate the relation between hedonic benefits and customer participation. Specifically, high perceived control will enhance the effect of hedonic benefits on customer participation, and low perceived control will diminish the effect*.

## 3. Method

### 3.1. Sample and Data Collection

For this study, we selected the electronics manufacturing industry because compared with other manufacturing companies, electronics companies are more commonly identified with customer-driven innovation, and they promote innovation through cooperation with suppliers and customers [48]. In addition, most electronic product enterprises establish an online exchange platform. Most online communities are active, and customers can participate in a wide range of activities.

We collected data from customers of online virtual platforms such as Millet, Huawei, ZTE, Meizu, Samsung, Apple, Nokia, Vivo HTC, and OPPO. Through online virtual communities, forums, and other online virtual platforms, the target enterprises issue questionnaires and collect data; examples include Huawei’s fans forum, Millet’s official community forum, MIUI’s official forum, and the Apple Forum. The data were collected from November 2015 to January 2016, a period of three months. During data collection, we distributed 575 questionnaires to the focal employees. We received 517 valid employee questionnaires, yielding a response rate of 89.9%.

First, we analyzed basic information, such as gender, age, education level, occupation, times, frequency of participation, and type of enterprise. The results showed that the gender distribution of respondents was relatively balanced. Almost all the respondents were 19–29 years old and had at least an undergraduate degree. In addition, more than half of them had been a part of their virtual community for over six months, and one-third participated at least once a week. Table 1 shows the distribution of the sample’s statistical indicators.

### 3.2. Variable Measurement

To ensure the accuracy of the questionnaire, we used the translation–back translation procedure. A bilingual professor translated the original version of the questionnaire into Chinese, and two PhD candidates translated the Chinese back into English. After the translation, we invited two employees to check whether the items were clear and accurate. Lastly, we established the scales, which included learning benefits, integrative benefits, hedonic benefits, perceived ease of use, customer participation, and the control variables. All measures used the same response scale, ranging from 1 (strongly disagree) to 5 (strongly agree).

Customer participation antecedents—these were divided into three dimensions, namely, learning benefits, integrative benefits, and hedonic benefits. For learning benefits, we used Zhang et al.’s four items [25], whose Cronbach’s alpha was 0.84. Sample items included “My interactions on the exchange platform enhance my knowledge about the products and their usage”. For integrative benefits, we used items from social integrative and personal integrative benefits [7].

We found that, after factor analysis, these two variables could synthesize a factor whose Cronbach’s alpha of integrative benefits was 0.92. Sample items included “My interactions on the exchange platform enhance my status/reputation as product expert in the community” and “My interactions on the exchange platform expand my personal/social network”. For hedonic benefits, we chose four items from Nambisan and Baron’s recently developed validated scale [7], whose Cronbach’s alpha was 0.88. Sample items included “I spend some enjoyable and relaxing time from the interactions on the exchange platform”.

Perceived ease of use—we adapted items used in the questionnaire from prior studies to operationalize the model constructs [34,38]. We used five items, namely, “I think it is very convenient to register as a member of the exchange platform”, “The platform plates are clear and reasonable and easy for me to participate in various activities”, “I can very easily and quickly post a message or thread back without going through the complex approval”, “I am able to communicate with anyone I want to and exchange ideas, without interference, such as from ranked members, or other restrictions”, and “I could easily apply to participate in the exchange platform events, and sufficient number of places are available”. Cronbach’s alpha was 0.84.

Perceived control—we measured perceived control based on the scale developed by Hui and Bateson [45]. It had four items to measure perceived control, namely, “My participation can help the company to grasp the customer’s needs”, “In the process of participation, I feel that everything is under my control”, “My participation to a certain extent can help the company to better grasp the quality of products and services”, and “I feel that my thoughts are valued in the process of participating”. Cronbach’s alpha was 0.88.

Customer participation. We measured customer participation using a scale used by Wang and Fesenmaier [21]. It had four items to measure customer participation. However, after factor analysis, we found that one of the items did not match the others. Hence, we selected three items to measure customer participation, which were “I lack strong social ties to the group, and seldom contribute to the community, the company and the products”, “I maintain somewhat strong social ties with the group, and sometimes contribute to the community, the company and the products”, “I maintain strong social ties with the group, I am enthusiastic about community activities and I contribute to the community, the company and the products often”. Cronbach’s alpha was 0.72.

Control variables—according to previous studies on customer participation motivation and customer participation behavior, customers’ gender, education level, tenure, gender, and times can influence customer participation behavior [7,19,25,27]. We controlled for organization effect and employees’ age and education because of their potential effects on employee behavior [49]. Therefore, this study selected gender, education level, tenure, gender, and times as control variables. Gender was a categorical variable (male = 1, female = 0).

### 3.3. Reliability and Validity Test

To ensure the quality of the collected data and enhance its reliability, this study strictly controlled the processes of questionnaire design, distribution and collection, and data entry. First, scales used in this survey were relatively mature scales, ensuring the quality and feasibility of the questionnaire. Second, respondents were from different regions and age groups and represented 14 enterprises in China. After collecting the questionnaires, we collated the data. Partially ineffective and incomplete questionnaires were excluded. Finally, all the qualified data were collected and analyzed.

This study used Cronbach’s α to describe the internal consistency of the questionnaire. The results showed that value of Cronbach’s α exceeded 0.7, indicating that the questionnaire had good consistency. The structural validity of the questionnaire was tested by factor analysis. Analysis showed that the cumulative variance contribution rate of each variable was more than 60%, which showed the questionnaire structure had good validity.

## 4. Data Analysis and Results

### 4.1. Correlation Analysis of Sample Variables

In this study, hierarchical regression analysis was used to analyze the data. Table 2 describes the mean, standard deviation, and correlation coefficients of the variables. Consistent with our arguments, learning benefits were positively associated with customer participation (r = 0.31, *P* < 0.01). Integrative benefits had a positive relationship with customer participation (r = 0.59, *P* < 0.01). Hedonic benefits were positively associated with customer participation (r = 0.30, *P* < 0.01). Perceived ease of use was positively associated with customer participation (r = 0.52, *P* < 0.01). These results provided preliminary evidence to support our hypotheses.

### 4.2. Hypothesis Test

We adopted hierarchical regression analysis to test Hypotheses 1–5.

It can be seen from Table 3, which shows a positive relationship between learning benefits and customer participation, that Hypothesis 1 was supported (β = 0.13, *p* < 0.01). Table 3, which shows a positive relationship between integrative benefits and customer participation, also supported Hypothesis 2 (β = 0.56, *p* < 0.01). The table shows that there was a positive relationship between hedonic benefits and customer participation, as well as a positive relationship between perceived ease of use and customer participation, indicating that Hypothesis 3 (β = 0.29, *p* < 0.01) and Hypothesis 4 (β = 0.38, *p* < 0.01) were supported.

Next, we tested the moderating effect of perceived ease of use. The results of the moderating effect of convenience participation on customer participation motivation and participation are shown in Table 3. There were three models. Model 1 was the basic model, including independent variables and control variables. Model 2 contained independent variables, control variables, and moderator variables. Model 3 contained all variables and the interaction of independent variables and moderator variables.

Table 3 also shows the results of the moderated hierarchical regression analyses for Hypothesis 5. The control variables were entered into the first step of the regression equation (Model 1 in Table 3). In the second step (Model 2 in Table 3), we added the independent variable into the regression equation based on the first step. In the third step (Model 3 in Table 3), we added the moderating variable and the interaction of the independent variable and moderating variable together into the regression based on the second step. As shown in Model 3, the coefficient of the interaction terms of learning benefits and perceived ease of use was negatively and not significantly related to customer participation (β = −0.01, *p* > 0.01, Model 3 in Table 3). Hypothesis 5a was not supported. The coefficient of the interaction of integrative benefits and perceived ease of use was positively and significantly related to customer participation (β = 0.08, *p* < 0.01, Model 3 in Table 3). Hypothesis 5b was supported. The coefficient of the interaction of hedonic benefits and perceived ease of use was positively and not significantly related to customer participation (β = 0.01, *p* > 0.01, Model 3 in Table 3). Hypothesis 5c was not supported.

To help interpret the moderating effects, we plotted the interaction effects. As shown in Figure 2, the positive relationship between integrative benefits and customer participation was much more distinct in high perceived ease of use rather than in low perceived ease of use.

As shown in Table 4, the coefficient of the interaction of learning benefits and perceived control was negative and not significantly related to customer participation (β = −0.05, *p* > 0.01, Model 3 in Table 4).

Hypothesis 6a was not supported. The coefficient of the interaction of integrative benefits and perceived control was negative and not significantly related to customer participation (β = 0.04, *p* > 0.01, Model 3 in Table 4). Hypothesis 6b was not supported. The coefficient of the interaction of hedonic benefits and perceived control was positively and significantly related to customer participation (β = 0.70, *p* < 0.01, Model 3 in Table 4). Hypothesis 6b was supported.

To help interpret the moderating effects, we plotted the interaction effects. As shown in Figure 3, the positive relationship between hedonic benefits and customer participation was much more distinct in high perceived control rather than in low perceived control.

## 5. Discussion 

From the perspective of social exchange theory, this study used survey data from 517 respondents to explore the relationship between customer participation antecedents and participation as well as the moderating effect of perceived ease of use on this relationship. The main conclusions are as follows.

The empirical results show that learning benefits positively relate to customer participation. Previous studies have stated that learning benefits have a significant, positive impact on customer participation [7,25], and the results of this study are consistent with that finding. An individual’s cognitive factor has a significant impact on their behavior [24]. According to the social exchange theory, social interaction is essentially a two-way process of reciprocity [17]. When customers have cognitive needs related to the company and its product, they may consider that participation in new product development will yield them benefits. When they interact online, they expect to access more information about the product [20]. Therefore, they may participate in new product development for obtaining certain products, price concessions or economic linking, and for receiving information that can offer preferential prices [50].

Integrative benefits have a positive effect on customer participation. Customers participate to gain some type of benefit, and satisfaction stemming from these benefits will stimulate further customer participation [7,21,25,27]. To gain recognition and position in a community, customers will participate in a wide range of innovation activities [31,32], which further promote their participation.

Hedonic benefits have a positive impact on customer participation. Some scholars have shown that hedonic benefits have a significant, positive impact on customer participation [7,21,27]. This study argues that virtual communities have different hedonic benefit characteristics. Customers prefer, or even enjoy, co-creation. On participation in new product development, customers will experience a sense of pleasure, enjoyment, freshness, or joy because they participate in producing innovation, in the selection process, and in designing and creating tasks and other stages [33]. The experience they gain directly activates a strong level of fantasy and joy [51]; this psychological feedback promotes customer participation [52].

Perceived ease of use has a positive effect on customer participation. Many scholars have pointed out that perceived ease of use plays an important role in the use of new technologies by customers [53]. From this study’s perspective, the interface style and interactivity of virtual communities affect customers’ perceived ease of use, which is a part of perceived ease of use [43]. Interaction can deepen interpersonal relationships among members, and convenient user access can help participants to maintain more and closer ties. According to the social exchange theory, exchange is easier when it is rational and equitable [42]. When customers consider participation in new product development easy, they are more willing to participate in a wide scope of development activities.

Perceived ease of use plays a positive, moderating role in the relationship between integrative benefits and customer participation. Customers participate in innovation activities for individual status, authority, reputation, and satisfaction. Perceived ease of use effectively promotes problem solving. A convenient virtual community enables customers to maintain relationships and deepen ties, since they are likely to easily meet like-minded people [21]. This deepens the interaction between customers themselves and between customers and enterprises. According to the social exchange theory, an exchange is more likely to occur when the exchange environment is favorable. As show in Figure 2, high perceived ease of use will enhance the effect of integrative benefits and customer participation.

Perceived control moderates the relationship between hedonic benefits and customer participation. Specifically, high perceived control will enhance the effect of hedonic benefits on customer participation, and low perceived control will diminish the effect. Customers with high levels of perceived control believe that their participation will be more meaningful [13] and, hence, will accept products more favorably [23] than those with low levels of perceived control. As shown in Figure 3, high perceived control will enhance the effect of hedonic benefits and customer participation.

This study also provides managerial implications for manufacturing enterprises. First, enterprises should pay attention to expected benefits of customers, particularly learning benefits, integrative benefits, and hedonic benefits. Second, enterprises should take full advantage of the internet and provide an interactive platform for active communication. Third, they should try to improve perceived ease of use in virtual communities and strengthen perceptions of the interface style, operation, and interaction to improve convenience and promote customer participation. Finally, manufacturers should regularly conduct activities for providing customers with a rich, colorful life in the community.

## 6. Research Limitations and Future Perspectives

Although this study has arrived at some encouraging conclusions, it has some shortcomings. First, customer participation in this study focuses on manufacturing enterprises and, in particular, on the electronics industry, within the context of virtual communities. It is necessary to study customer participation within different contexts and to examine different social media types in the future to ensure that the results can be validated and applied more widely. Second, the survey data show that most of the participants are students and youth groups. The reason may be that the consumer groups of the electronic products of the researched companies are mostly young people. A future study can investigate the participation of other age groups in the manufacturing industry or other age groups in the electronics industry.

## 7. Conclusions

This research aims to answer the question that why customers participate in innovation and explores effects of customer participation antecedents (learning benefits, integrative benefits and hedonic benefits) on customer participation. An empirical study of 517 Chinese participants confirmed most of the proposed hypotheses. The results indicate that learning benefits, integrative benefits and hedonic benefits have positive effects on customer participation, and perceived control functions as a moderator during these three paths, while perceived ease of use only moderates the relationship between integrative benefits and customer participation. Our research findings generate several implications indicated above.

## Figures and Tables

**Figure 1 ijerph-16-04998-f001:**
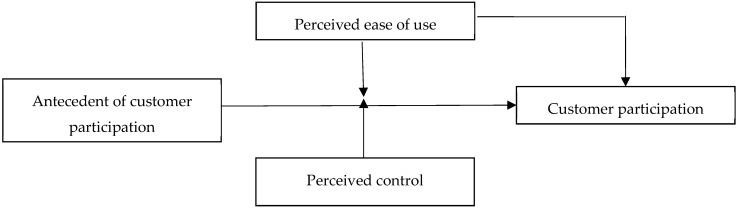
Theoretical model.

**Figure 2 ijerph-16-04998-f002:**
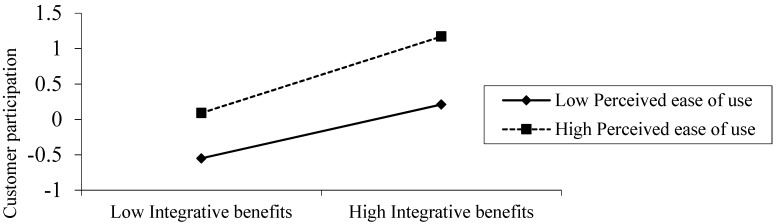
The moderating effects of perceived ease of use.

**Figure 3 ijerph-16-04998-f003:**
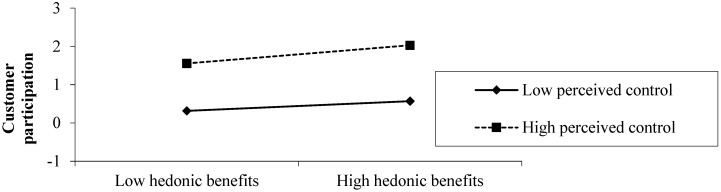
The moderating effects of perceived control.

**Table 1 ijerph-16-04998-t001:** Distribution of sample statistical indicators (*N* = 517).

Control Variables		Number	Control Variables		Number
Tenure	<6 months	214	Gender	Male	315
	6–12 months	88		Female	202
	13–24 months	158	Position	Students	387
	25–48 months	57		Workers	109
Times	<5	354		Underemployed	12
	5–10	98		Others	9
	11–20	34	Education	High School	21
	>20	31		Junior College	37
Age	<18	17		Bachelor	244
	19–29	483		Master	209
	30–39	13		Ph.D. candidate	6
	>40	4			

**Table 2 ijerph-16-04998-t002:** Mean, standard deviation, and correlation coefficient matrix.

Variable	Mean	SD	1	2	3	4	5	6	7	8	9
1. Education	3.27	0.78									
2. Tenure	2.11	1.07	−0.17 **								
3. Gender	0.61	0.49	0.21 **	−0.18 **							
4. Times	1.50	0.86	−0.27 **	0.36 **	−0.17 **						
5. Learning Benefits	3.84	0.79	−0.10 *	0.14 **	0.07	0.20 **					
6. Integrative benefits	3.43	0.88	−0.14 **	0.10 *	−0.05	0.34 **	0.56 **				
7. Hedonic benefits	3.53	0.90	−0.12 **	0.08	−0.02	0.29 **	0.59 **	0.69 **			
8. Perceived ease of use	3.63	0.75	−0.15 **	0.13 **	−0.02	0.25 **	0.54 **	0.49 **	0.48 **		
9. Perceived control	3.16	0.88	−0.11 *	0.04	−0.09	0.26 **	0.27 **	0.60 **	0.44 **	0.48 **	
10. Customer participation	3.30	0.76	−0.18 **	0.04	−0.12 **	0.25 **	0.35 **	0. 63 **	0.51 **	0.53 **	0.69 **

Note: *n* = 517. * *p* < 0.05, ** *p* < 0.01.

**Table 3 ijerph-16-04998-t003:** Moderating effect of perceived ease of use between antecedents of customer participation and customer participation.

	Model 1				Model 2				Model 3			
Independent Variable	Coefficient	t	Sig.	VIF	Coefficient	t	Sig.	VIF	Coefficient	t	Sig.	VIF
Education (C1)	−0.17	−2.21	0.03	1.12	−0.08	−1.81	0.07	1.13	−0.07	−1.60	0.11	1.34
Tenure (C2)	−0.10	−0.42	0.67	1.19	−0.02	−0.61	0.54	1.19	−0.02	−0.47	0.64	1.20
Gender (C3)	−0.14	−2.03	0.04	1.09	−0.14	02.09	0.04	1.09	−0.13	−1.96	0.05	1.10
Times (C4)	0.07	1.55	0.12	1.37	−0.05	1.15	0.25	1.37	0.04	0.04	0.31	1.38
Learning benefits (X_1_)	0.13	3.83	0.00	1.03	0.02	0.53	0.60	1.26	0.01	0.36	0.72	1.33
Integrative benefits (X_2_)	0.56	16.37	0.00	1.11	0.47	13.59	0.000	1.26	0.46	13.24	0.00	1.28
Hedonic benefits (X_3_)	0.29	8.65	0.00	1.04	0.21	6.40	0.00	1.15	0.20	5.94	0.00	1.20
Perceived ease of use (M)					0.38	7.33	0.00	155	0.40	7.70	0.00	1.65
X_1_ * M									−0. 01	−0.73	0.47	1.15
X_2_ * M									0.08	2.59	0.00	1.16
X_3_ * M									0.01	0.31	0.76	1.17
R^2^	0.47				0.52				0.53			
ΔR^2^	3.55				0.51				0.07			
ΔF	16.28				113.08				53.77			
Sig. ΔF	0.000				0.000				0.06			
Sig. of Model	0.000				0.000				0.000			

Note: *n* = 517. VIF = Variance Inflation Factor. * = Multiplication sign.

**Table 4 ijerph-16-04998-t004:** Moderating effect of perceived control between antecedents of customer participation and customer participation.

	Model 1				Model 2				Model 3			
Independent Variable	Coefficient	t	Sig.	VIF	Coefficient	t	Sig.	VIF	Coefficient	t	Sig.	VIF
Education (C1)	−0.10	−2.50	0.03	1.12	−0.09	−2.45	0.02	1.12	−0.10	−2.53	0.01	1.13
Tenure (C2)	−0.14	−0.43	0.67	1.19	−0.03	0.10	0.92	1.20	0.00	−0.15	0.88	1.20
Gender (C3)	−0.14	−2.03	0.43	1.09	−0.10	−1.64	0.10	1.10	−0.09	−1.55	0.12	1.11
Times (4)	0.07	1.55	0.12	1.37	−0.03	0.86	0.39	1.37	0.03	0.69	0.49	1.38
Learning benefits (X_1_)	0.13	3.83	0.00	1.03	0.10	3.39	0.01	1.04	0.23	2.66	0.01	9.83
Integrative benefits (X_2_)	0.56	16.37	0.00	1.11	0.32	8.98	0.00	1.56	0.43	5.18	0.00	8.78
Hedonic benefits (X_3_)	0.29	8.65	0.00	1.04	0.18	6.16	0.00	1.12	−0.02	−0.20	0.84	10.97
Perceived control (M)					0.51	12.60	0.00	1.59	0.52	12.90	0.00	1.60
X_1_ * M									−0.05	−1.70	0.10	9.94
X_2_ * M									−0.04	−1.40	0.16	8.53
X_3_ * M									−0.70	2.31	0.02	10.86
R^2^	0.47				0.59				0.60			
adjusted R^2^	0.46				0.59				0.59			
ΔF	113.08				158.78				3.20			
ΔR^2^	0.36				0.13				0.01			
Sig. of Model	0.000				0.000				0.000			

Note: *n* = 517. VIF = Variance Inflation Factor. * = Multiplication sign.

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
