# Peer review of "Customers as Co-Creators: Antecedents of Customer Participation in Online Virtual Communities"

_ijerph, 2019, doi:10.3390/ijerph16244998_

Round 1

Reviewer 1 Report

The work, Customer as Co-creator: Antecedents of Customer Participation in Online Virtual Communities, is well written. The scientific content is good, and presented
results are rather convincing.

I suggest to improve the English because some sentences are not always clear and to upgrade the literature survey with references be of the last five years. Also, I suggest removing parts that quote tables and images from the conclusion and leave them in discussions and results.

I congratulate the authors and look forward to the results of their future research in this fruitful field.

Author Response

For Reviewer #1:

Question 1:The work, Customer as Co-creator: Antecedents of Customer Participation in Online Virtual Communities, is well written. The scientific content is good, and presented results are rather convincing.

Answer:Thank you very much for your review.

Question 2:I suggest to improve the English because some sentences are not always clear and to upgrade the literature survey with references be of the last five years. Also, I suggest removing parts that quote tables and images from the conclusion and leave them in discussions and results.

Answer:Thanks for your advice. First, we find an English polishing company to deal with the language. We have rechecked the writing again, including grammar, sentence structure and try to express it in a better way. Second, we have supplemented some literature in recent years.

Zhao, J., Tao, J., & Xiong, G. Online Brand Community Climate, Psychological Capital, and Customer Value Cocreation. Social Behavior and Personality: An International Journal, 2019, 47(3), 1–14. Gallan, A. S., Cheryl, B. J., Brown, S.W. & Bitner, M. J. “Customer positivity and participation in services: an empirical test in a health care context”, Journal of the Academy of Marketing Science, 2013, 41(3), 338-356. Gruner R L, Homburg, C., & Lukas, B. A. Firm-hosted online brand communities and new product success. Journal of the Academy of Marketing Science, 2014, 42(1):29-48. Hansen, J. M., Saridakis, G., & Benson, V. Risk, trust, and the interaction of perceived ease of use and behavioral control in predicting consumers’ use of social media for transactions. Computers in Human Behavior, 2018, 80:197-206.

Question 3:I congratulate the authors and look forward to the results of their future research in this fruitful field.

Answer:Thank you so much.

Reviewer 2 Report

Authors present a study where they show that customer participation in new product development is positively correlated with learning, integrative and hedonic benefits.

The paper is well written in sufficient detail. However, I find the following two major issues:

Dataset is too small to make a decisive judgment. There were only 517 responses from 10 companies (average 51.7 responses per company). Participants are company employees, hence their opinion/survey is biased and cannot reflect the opinion of the public.

Minor issues/typos:

The abstract does not summaries the work well. There could be numbers, p-values etc to give more inside details in the abstract. Line 25 – Internet should be written in lower case. Line 62: replace “little” with a better word such as lack of. Line 277 “commonly identified with customer-driven” should read as “commonly identified as customer-driven” Line 291 replace the word “majority” with some scientific numbers such as IQR. Table 2: * and ** are not defined. Lines 370 and 372 replace predicts with better words such as shows/demonstrates. Line 373 “And in the meanwhile” does not sound good here Line 404 – “perceived control is negatively” does not make sense.

Author Response

For Reviewer #2:

Thank you very much for your valuable advice.

Question 1:Dataset is too small to make a decisive judgment. There were only 517 responses from 10 companies (average 51.7 responses per company). Participants are company employees, hence their opinion/survey is biased and cannot reflect the opinion of the public.

Answer:This study does not focus on all industries.According to the research topic of this study, this study chooses manufacturing industry as the object. The specific reasons and descriptions of the samples are as follows:This paper selected the electronics manufacturing industry because compared to other manufacturing companies, electronics companies are more commonly identified with customer-driven innovation, and they promote innovation through cooperation with suppliers and customers [48]. At the same time, majority of electronic products enterprises establish an online exchange platform. Most online communities are active, and customers can participate in a wide range of activities.

This paper collect data from the online virtual platform customers of Millet, HUAWEI, ZTE, Meizu, Samsung, Apple, NOKIA, VIVO, HTC, OPPO and other enterprises. Through online virtual communities, forums and other online virtual platforms, the target enterprises issue questionnaires and collect data; examples include HUAWEI’s fans forum(http://cn.club.vmall.com/), Millet’s official community forum (http://bbs.xiaomi.cn/), MIUI’s official forum(http://www.miui.com/forum.php), and the Apple Forum(http://bbs.app111.com/). The data are collected from November 2015 to January 2016, a period of three months. During data collection, we distribute 575 questionnaires to the focal employees. We receive 517 valid employee questionnaires, for a response rate of 89.9%.

Question 2:The abstract does not summaries the work well. There could be numbers, p-values etc to give more inside details in the abstract. Line 25 – Internet should be written in lower case. Line 62: replace “little” with a better word such as lack of. Line 277 “commonly identified with customer-driven” should read as “commonly identified as customer-driven” Line 291 replace the word “majority” with some scientific numbers such as IQR. Table 2: * and ** are not defined. Lines 370 and 372 replace predicts with better words such as shows/demonstrates. Line 373 “And in the meanwhile” does not sound good here Line 404 – “perceived control is negatively” does not make sense.

Answer:Thanks for your advice. The abstract has been improved.The revised content is as follows:

Abstract: The development of internet technology and the popularity of smartphones has been gradually affecting people’s daily life, thus causing subtle changes to their health. Manufacturing companies are increasingly establishing virtual communities to motivate customers to participate in new product development. However, the reasons that customers participate in the innovation process and the timing of participation remain under-researched. Hence, using data on 517 customers of 14 manufacturing enterprises, we investigate the reasons behind such participation and the moderating role of perceived ease of use and perceived control based on the social exchange theory. Results show that learning benefits, integrative benefits and hedonic benefits have positive effects on participation. Perceived ease of use strengthens the positive relationship between integrative benefits and customer participation. Perceived control strengthens the positive relationship between hedonic benefits and customer participation. Theoretical implications and managerial practices are also discussed.

We find an English polishing company to deal with the language. We have rechecked the writing again, including grammar, sentence structure and try to express it in a better way.

Line 62, replace “little” with as lack of

Line 277 has been changed into:commonly identified as customer-driven

Line 291 has been changed into: The respondents are almost between 19 and 29 years old and have an undergraduate degree or above. In addition, more than half of the respondents had been a part of their virtual community for over six months, and 1/3 customers participate at least once a week. Table 1 also shows the statistical indicators distribution of sample.

We add the fowling content after table 2: Note: n = 517. *p < .05, **p < .01.

Lines 370 and 372 replace predicts with shows.

Line 373 “And in the meanwhile” is deleted.

Line 400-404 has been changed into: As shown in Table 4, the coefficient of the interaction of learning benefits and perceived control is negative and not significantly related to customer participation (β = -0.05, p > .01, Model 3 in Table 4).

Hypothesis 6a is not supported. The coefficient of the interaction of integrative benefits and perceived control is negative and not significantly related to customer participation (β = 0.04, p >.01, Model 3 in Table 4). Hypothesis 6b is not supported.

Reviewer 3 Report

The authors use familiar frames (Uses & Gratifications Theory, Social Exchange Theory) in a meaningful way to extend prior research on why consumers use technology to why they decide whether or not to participate in service-related online virtual communities.

Although the survey response rate is high, I wonder how many total virtual platform customers you had the ability to canvass, given your multiple sources of contact/multiple virtual platforms?  There is a possibility that your respondents could be either those who highly value the platform/product or those who highly dislike it.

I would suggest more extensive development of the research limitations and future perspectives.  See the note above, and consider the fact that you have a highly homogeneous pool of respondents.  What insights into the theories utilized can you 'tease out' in this final section?

There are a number of grammatical errors that should be addressed in a final manuscript submission prior to publication; this, as well as the abbreviated final section of the article, results in my lower rating for quality of presentation (the actual summaries/presentation of data and results are well done).  Whether done by the authors or editors, there is a need to address missing words in phrases, errors in subject/verb agreement, and incomplete sentences.  I have given specific examples to the editors.

Author Response

For Reviewer #3:

Thank you very much for your valuable advice.

Question 1:Although the survey response rate is high, I wonder how many total virtual platform customers you had the ability to canvass, given your multiple sources of contact/multiple virtual platforms? There is a possibility that your respondents could be either those who highly value the platform/product or those who highly dislike it.

Answer:Thanks for your advice. We add some content in limitation part, the content is as follows: Second, the survey data show that most of the participants are students and youth groups. The reason may be that the consumer groups of the electronic products of the researched companies are mostly young people. A future study can investigate the participation of other age groups in the manufacturing industry or other age groups in the electronics industry.

Question 2: I would suggest more extensive development of the research limitations and future perspectives.  See the note above, and consider the fact that you have a highly homogeneous pool of respondents.  What insights into the theories utilized can you 'tease out' in this final section?

The part of Research Limitations and Future Perspectives is changed into:

Although this study has arrived at some encouraging conclusions, it has some shortcomings. First, customer participation in this study focuses on manufacturing enterprises, and, in particular, on the electronics industry, within the context of virtual communities. It is necessary to study customer participation within different contexts and to examine different social media types in the future to ensure that the results can be validated and applied more widely. Second, the survey data show that most of the participants are students and youth groups. The reason may be that the consumer groups of the electronic products of the researched companies are mostly young people. A future study can investigate the participation of other age groups in the manufacturing industry or other age groups in the electronics industry.

Question 3: There are a number of grammatical errors that should be addressed in a final manuscript submission prior to publication; this, as well as the abbreviated final section of the article, results in my lower rating for quality of presentation (the actual summaries/presentation of data and results are well done).  Whether done by the authors or editors, there is a need to address missing words in phrases, errors in subject/verb agreement, and incomplete sentences.  I have given specific examples to the editors.

Answer:Thanks for your advice. We find an English polishing company to deal with the language. We have rechecked the writing again, including grammar, sentence structure and try to express it in a better way.

Round 2

Reviewer 3 Report

All concerns have been addressed well within this revised manuscript.